# Single Nucleotide Polymorphism of TWIST2 May Be a Modifier for the Association between High-Density Lipoprotein Cholesterol and Blood Lead (Pb) Level

**DOI:** 10.3390/ijerph19031352

**Published:** 2022-01-26

**Authors:** Chen-Cheng Yang, Chia-Yen Dai, Kuei-Hau Luo, Kuo-Wei Lee, Cheng-Hang Wu, Chih-Hsing Hung, Hung-Yi Chuang, Chao-Hung Kuo

**Affiliations:** 1Graduate Institute of Medicine, College of Medicine, Kaohsiung Medical University, Kaohsiung City 807, Taiwan; u106800001@kmu.edu.tw (C.-C.Y.); u107800007@kmu.edu.tw (K.-H.L.); 2Department of Occupational Medicine, Kaohsiung Municipal Siaogang Hospital, Kaohsiung Medical University, Kaohsiung City 812, Taiwan; 3Department of Occupational and Environmental Medicine, Kaohsiung Medical University Hospital, Kaohsiung Medical University, Kaohsiung City 807, Taiwan; daichiayen@gmail.com; 4Department of Neurology, Kaohsiung Municipal Siaogang Hospital, Kaohsiung Medical University, Kaohsiung City 812, Taiwan; 1030399@kmuh.org.tw; 5Department of Family Medicine, Kaohsiung Municipal Siaogang Hospital, Kaohsiung Medical University, Kaohsiung City 812, Taiwan; 1010352@kmuh.org.tw; 6Environmental and Occupational Medicine Center, Kaohsiung Municipal Siaogang Hospital, Kaohsiung Medical University, Kaohsiung City 812, Taiwan; pedhung@gmail.com; 7Department of Public Health and Environmental Medicine, Research Center for Environmental Medicine, Kaohsiung Medical University, Kaohsiung City 807, Taiwan; 8Department of Internal Medicine, Kaohsiung Municipal Siaogang Hospital, Kaohsiung Medical University, Kaohsiung City 812, Taiwan; jhkao@kmu.edu.tw

**Keywords:** lead (Pb), high-density lipoprotein cholesterol (HDL-C), single nucleotide polymorphism, genome-wide association study, Taiwan biobank

## Abstract

The association between lead (Pb) exposure and lower high-density lipoprotein cholesterol (HDL-C) was reported; however, the mechanism was unclear. Our purpose was to investigate the association of Pb, lipid profile, and to study the associated SNPs using a genome-wide association study (GWAS). A total of 511 participants were recruited to check blood Pb levels, lipid profile, and genotypes with Taiwan Biobank version 2.0 (TWB2). Our main result shows that HDL-C was significantly negatively associated with blood Pb levels, adjusted for gender, body mass index (BMI), and potential confounders. In addition, via the TWB2 GWAS, only two SNPs were found, including rs150813626 (single-nucleotide variation in the TWIST2 gene on chromosome 2), and rs1983079 (unclear SNP on chromosome 3). Compared to the rs150813626 GG carriers, the AA and AG carriers were significantly and negatively associated with HDL-C. We analyzed the interaction of rs150813626 SNP and blood Pb, and the HDL-C was consistently and negatively associated with blood Pb, male, BMI, and the rs150813626 AA and AG carriers. Moreover, the rs150813626 AA and blood Pb interaction was significantly and positively associated with HDL-C. In conclusion, the SNPs rs150813626 and rs1983079 were significantly associated with HDL-C in Pb-exposed workers. Furthermore, the interaction of rs150813626 AA and blood Pb had a positive influence on HDL-C. TWIST may inhibit osteoblast maturation, which might relate to bone Pb deposition and calcium metabolism. The mechanism needs more investigation in the future.

## 1. Introduction

### 1.1. Lead Toxicity

Lead (Pb) toxicity is an important issue in occupational and environmental health. It causes renal toxicity [1,2], hemopoietic system disorder [3], and intelligence [4] and mental retardation [5]. In Taiwan, leaded gasoline had been prohibited since 2000, and the blood Pb levels of the community population decreased gradually as time went by [6]. However, occupational Pb exposure, such as for workers in a metal recycle factory or battery factory, is still a high risk.

### 1.2. Lipid Profile and Lead Exposure

Previous studies have shown that Pb exposure is associated with cardiovascular risk factors, including dyslipidemia [7,8,9]. Gategonova et al. found that Pb in exposed workers is associated with elevated triglyceride (TG), total cholesterol (TC), and low-density lipoprotein cholesterol (LDL-C) but decreased high-density lipoprotein cholesterol (HDL-C) [8]. In a study involving 590 preschool children, Lu et al. showed that higher blood Pb is associated with higher TG but lower HDL-C [9]. On the other hand, in a cross-sectional study with 479 male workers in a battery recycling plant, Ghiasvand et al. showed no association between high blood Pb levels and lipid profiles, including TC, TG, HDL-C, and LDL-C [10]. In the 10–18-year-old offspring of mothers who had prenatal Pb exposure, Liu et al. found that boys of those mothers’ blood Pb levels ≥ 5 µg/dL during pregnancy had significantly lower TC, HDL-C, and LDL-C [11]. From the above studies, the relationship between Pb exposure and the lipid profile is still controversial. Hence, more studies about the relationship are required.

### 1.3. Single Nucleotide Polymorphism (SNP), Lipid Profile, and Lead Exposure

Among individuals, the most common style of DNA variation is the replacement of one single nucleotide for another, which is termed single nucleotide polymorphism (SNP). In whole genome sequences, SNPs are anticipated to happen at an incidence of around one in 1000 base pairs [12]. SNPs can be found among individuals, play important role in gene expression [13], influence the stability of RNA [14,15], and affect translational efficiency [16,17]. These impacts may influence the phenotypical manifestation of diseases [18] or personal characteristics, including body height [19], body weight [20], body mass index (BMI) [21], or skin trait variation [22]. On the other hand, only limited studies about whether SNPs influence the association between Pb exposure and lipid profilers have been conducted until now. Li et al. suggested that particularly in the RR carriers of Q192R, increased blood Pb levels are significantly and negatively associated with paraoxonase 1 activity (PON1), which represents reduced protection against LDL-C oxidation [23]. Kamal et al. declared that Pb-exposed workers have a significant positive association with TC, TG, and LDL-C but a negative association with HDL-C. Moreover, blood Pb levels were significantly negatively associated with serum paraoxonase activity, and the carriers of QQ and QR of the Q192R of the paraoxonase genotype between control and exposed participants were significantly different [24]. However, the above studies did not investigate how the whole genome genetic polymorphisms affect the association between lipid profiles and Pb exposure. Hence, further investigation is needed.

### 1.4. Aim

Our purpose was to study the association between lipid profiles and Pb exposure, whether the association would be influenced by certain associated genetic polymorphisms, and to investigate the associated SNPs using a genome-wide association study (GWAS).

## 2. Materials and Methods

### 2.1. Participants and Health Examination

According to the Regulations of Labor Insurance Health Examination for the Prevention of Occupational Disease in Taiwan, workers exposed to Pb in the workplace should have health examinations regularly. The annual health examination includes a physical examination, blood Pb measurement, complete blood count (CBC), and liver and renal function examinations. Pb-exposed workers from a battery factory were initially recruited for the study. After informed consent was acquired, following the guidelines of the Kaohsiung Medical University Hospital Institutional Review Board for studies on human beings, blood samples were obtained for lipid profiles and genetic analysis (IRB number KMUIRB-E(I)-20190034 and date of approval:
5 March 2019). The Department of Laboratory Medicine in Kaohsiung Medical University Hospital oversaw all the biochemistry examinations, and the technicians were blinded to the purpose of the study and the exposure condition of the participants. In addition, a short questionnaire requesting information about their job title, medical and working history, medication use, smoking, and alcohol consumption was a component of the study.

### 2.2. Pb Level Measurement

We used a standard operation procedure to gather venous blood samples from all participants. The Zeeman-effect graphite furnace atomic absorption spectrometer (GF-AAS, Perkin-Elmer 5100 PC with an AS 60 autosampler, PerkinElmer, Inc., Waltham, MA, USA) was used for the measurement of the Pb levels in the whole blood samples. All laboratory consumables were standard commercial materials (Bio-Rad Laboratories, Inc., Hercules, California, United States). The coefficients of variation (CVs) of high (184.00–666.00 μg/L) and medium levels (60.00–183.99 μg/L) of blood Pb were less than 3%, and the CVs of low levels (1.91–59.99 μg/L) of blood Pb were less than 5%.

### 2.3. Taiwan Biobank (TWB) Version 2.0 and Genome-Wide Association Study (GWAS)

Taiwan Biobank (TWB) version 2.0 (C2-58 Axiom Genome-Wide TWB 2.0 Array) is based on the technology platform developed by Affymetrix in the United States. It used the Axiom Genome-Wide Array Plate system to select a total of 714,431 SNPs. The SNPs of TWB version 2.0 consist of (1) the data of TWB version 1.0 and the whole genome sequencing results of 1000 participants in TWB, for a total of 446,000 SNPs, (2) 105,000 clinically significant SNPs, (3) disease-related SNPs for Biobanks in various countries designed by Thermo Fisher Scientific over the years, (4) others related to drug reaction and drug metabolism, such as SNPs on such genes as major histocompatibility complex (MHC) and pharmacogenomic (PGX), and (5) SNPs that detect copy number variation (CNV) [25,26,27]. The genotypes of each participant were performed by National Center for Genome Medicine (NCGM) at Academic Sinica using the Axiom Genome-Wide TWB Array Plate with a total of 714,431 SNPs. We applied the TWB version 2.0 whole genome genotyping to establish genomic data and conducted a GWAS for the Pb-exposed workers. Because the participants were all Han Chinese, we did not conduct ethnicity analysis in the GWAS. However, subjects with a high missing rate of genotyping or an extreme heterozygosity rate were excluded for the quality control of genotype data. If the missing genotype rate was >5%, we excluded the participant. The formula (N-O)/N was applied to determine the mean heterozygosity rate, where N means the number of non-missing genotypes, and O means the observed number of homozygous genotypes. If the extreme heterozygosity rate was more than three standard deviations (SD) from the mean heterozygosity rate, this suggested the participant’s sample was inbreeding or contaminated, so participants with an extreme heterozygosity rate were excluded. SNPs were omitted if the *p*-value of the Hardy–Weinberg equilibrium (HWE) was less than 10^−6^. Moreover, an independent sample was defined as an identity-by-descent (IBD) of less than 0.1875; thus, we excluded individuals if their IBD was > 0.1875. All quality controls were conducted using PLINK software (version 1.9) [28] and R language version 3.5.2 software [29].

### 2.4. Statistics

The descriptive statistics with the means and dispersion were analyzed and expressed with continuous variables, including blood Pb level, age, BMI, TG, TC, LDL-C, and HDL-C. For categorical variables, including gender, smoking, and alcohol consumption, the numbers and proportions of these variables were applied. PLINK version 1.9 [28] and R version 3.5.2 software [29] were used for the genome-wide association study (GWAS), selecting the novel possible associated genes of lipid profiles. Furthermore, a regression model was used to assess the association of lipid levels and blood Pb levels with adjustment for novel associated SNPs and other potential confounders, including age, gender, BMI, smoking, and alcohol consumption. IBM-SPSS version 21 statistical software was utilized for the descriptive analysis and regression model, with *p*-values of less than 0.05 considered significant. However, the Bonferroni correction was used to correct the *p*-value of multiple testing. The corrected *p*-value is 0.05/number of SNPs.

## 3. Results

### 3.1. Demographic Characteristics

Table 1 reveals the demographic characteristics of the 511 participants, including 281 (55%) males and 230 (45%) females in the study. The means and SDs of age, body weight, body height, and BMI are 42.79 ± 10.48 years, 66.73 ± 14.15 kg, 163.33 ± 9.30 cm, and 24.90 ± 4.24 kg/m^2^, respectively. In addition, the mean and SD of the blood Pb level is 139.94 ± 111.11 ug/L. The mean and SD of the TG, TC, HDL-C, and LDL-C are 137.95 ± 144.02 mg/dL, 205.31 ± 38.38 mg/dL, 48.45 ± 13.51 mg/dL, and 132.68 ± 32.98 mg/dL, respectively. Furthermore, 166 participants (32.5%) were smokers and 163 participants (31.9%) consumed alcohol.

### 3.2. Regression Model of Lipid Profiles

Table 2 shows the regression model of the lipid profile, blood Pb levels, and other potential confounders, including gender, age, BMI, smoking, and alcohol consumption. TG was significantly and positively associated with being male, BMI, and smoking, the β values of which were 31.65 (standard error (SE) 14.03, *p*-value 0.03), 7.65 (SE 1.48, *p*-value < 0.001), and 54.57 (SE 15.19, *p*-value < 0.001), respectively. Moreover, TC was significantly and positively associated with age, BMI, and smoking, the β values of which were 0.83 (SE 0.17, *p*-value < 0.001), 0.94 (SE 0.39, *p*-value 0.02), and 9.85 (SE 4.00, *p*-value 0.01), respectively. On the other hand, TC was significantly and negatively associated with being male, the β value of which was −9.89 (SE 3.70, *p*-value 0.01). Furthermore, HDL-C was significantly and negatively associated with blood Pb level, being male, and BMI, the β values of which were −0.025 (SE 0.005, *p*-value < 0.001), −6.041 (SE 1.23, *p*-value < 0.001), and −1.004 (SE 0.13, *p*-value < 0.001), respectively. LDL-C was significantly and positively associated with age, the β value of which was 0.62 (SE 0.15, *p*-value < 0.001).

### 3.3. Regression of the Association among HDL-C, Blood Pb, and Possible Associated SNPs

The GWAS was conducted on HDL-C to investigate the possible associated SNPs, the results of which are shown in Figure 1. Finally, after quality control of the genotype data, 289,659 of the total 714,431 SNPs were included in the GWAS; therefore, the corrected *p*-value was (0.05/289,659) = 2 × 10^−7^ and the suggested *p*-value < 1 × 10^−5^ was selected for testing. Above the line of -log (1 × 10^−5^), we found the possible SNPs that were associated with HDL-C were rs150813626 (single-nucleotide variation in the TWIST2 gene, an intron variant, on chromosome 2, chr2: 238,859,528 (GRCh38)) and rs1983079 (unclear SNP on chromosome 3, 3:114243087 (GRCh38)). Then we selected the two SNPs for multiple regression analyses.

Table 3 presents the three regression models of HDL-C, the two SNPs, and other potential confounders. Model 1 was the regression model of HDL-C, the blood Pb level, rs150813626 SNPs, and other potential confounders. HDL-C was significantly and negatively associated with blood Pb, being male, and BMI, the β values of which were −0.023 (SE 0.005, *p*-value < 0.001), −6.032 (SE 1.22, *p*-value < 0.001), and −1.014 (SE 0.13, *p*-value < 0.001), respectively. Compared to the rs150813626 GG carriers, the rs150813626 AA and rs150813626 AG carriers were significantly and negatively associated with HDL-C, the β values of which were −6.34 (SE 1.56, *p*-value < 0.001) and −2.44 (SE 1.20, *p*-value 0.042), respectively. In model 2, the interaction between the rs150813626 SNP and blood Pb level was integrated for further analysis. HDL-C was consistently negatively associated with blood Pb, being male, BMI, and the rs150813626 AA and AG carriers, the β values of which were −0.036 (SE 0.008, *p*-value < 0.001), −5.85 (SE 1.22, *p*-value < 0.001), −1.008 (SE 0.13, *p*-value < 0.001), −10.27 (SE 2.47, *p*-value < 0.001), and −4.90 (SE 1.89, *p*-value 0.010), respectively. Furthermore, the interaction between the rs150813626 AA allele and blood Pb level was significantly and positively associated with HDL-C, the β value of which was 0.027 (SE 0.013, *p*-value 0.036). Figure 2A shows the finding that the rs150813626 AA and rs150813626 AG carriers had a less steep slope for the association between the blood Pb level and HDL-C compared to the rs150813626 GG carrier.

In model 3 (Table 3), HDL-C was significantly and negatively associated with blood Pb level, being male, and BMI, the β values of which were −0.026 (SE 0.005, *p*-value < 0.001), −6.34 (SE 1.20, *p*-value < 0.001), and −0.94 (SE 0.13, *p*-value < 0.001), respectively. Compared to the rs1983079 GG carriers, HDL-C was positively associated with the rs1983079 AA and AG carriers, the β values of which were 8.028 (SE 1.67, *p*-value < 0.001) and 2.92 (SE 1.16, *p*-value 0.012), respectively. There was no interaction effect of the rs1983079 alleles and blood Pb level on HDL-C. Figure 2B shows the finding that the rs1983079 GG and AG carriers had a less steep slope for the association between the blood Pb level and HDL-C compared to the rs1983079 AA carrier.

## 4. Discussion

To our best knowledge, this might be the first study investigating the association between HDL-C and blood Pb levels using a GWAS on Pb-exposed workers in Taiwan. Genetic biochip technology and bioinformation have evolved in the past decade and have become more practical both in laboratories and in clinical situations. In this study, we discovered that HDL-C was significantly and negatively associated with blood Pb levels. Kristal-Boneh et al. showed that TC and HDL-C levels were higher in an occupational Pb-exposed group than in a non-Pb-exposed group [30]; however, the sample size was relatively small with 56 Pb-exposed participants versus 87 non-exposed participants, and the dose–response relationship of TC was only shown in the Pb-exposed workers [30]. Souza-Talarico et al. found that blood Pb levels were positively correlated with HDL-C in elder Brazilian participants [31]. Nevertheless, compared to our participants, the participants of Souza-Talarico’s study were relatively older, more of them were female, and their blood Pb levels were relatively low, ranging from 0.6 to 6.1 μg/dL (mean 2.1 μg/dL; standard deviation 0.9 μg/dL) [31]. Lin et al. found increased urinary Pb concentrations were positively associated with the prevalence of metabolic syndrome, especially in the diagnostic categories of waist size, BMI, and serum HDL-C of metabolic syndrome [32]. However, after adjusting for age and sex, the urinary Pb concentration was significantly negatively associated with HDL-C (adjusted β value −0.660, SE 0.203, *p*-value 0.001) [32]. On the other hand, Kamal et al. showed Pb exposure was positively correlated with TG, TC, and LDL-C but negatively correlated with HDL-C [24]. HDL-C gradually decreased as the blood Pb level increased, with a significant difference between the group whose blood Pb levels were between 40–59 ug/dL and the group whose blood Pb levels were greater than 60 ug/dL [24]. In 677 Pb-exposed workers, Yang et al. found a significantly negative association between HDL-C and blood Pb concentration [33]. In summary, our study confirmed that increased occupational Pb levels are significantly associated with decreased HDL-C, but the mechanism of how Pb toxicity influences HDL-C or lipid metabolism still requires further study.

Two potential associated SNPs on the novel loci influenced the HDL-C levels in the Pb-exposed workers. The SNP rs150813626, an intron variant, is located on chromosome 2, chr2: 238,859,528(GRCh38) and belongs to *TWIST2* family (TWIST2 located from chr2: 238,848,085 to 238,910,534, GRCh38) [34]. In 1987, Thisse et al. initially found that the transcription factor *TWIST* genes in Drosophila could be found in humans [35]. The *TWIST2* family, the official full name being *TWIST family basic helix-loop-helix transcription factor 2*, is crucial for the progression of bone development, embryonic skeletal muscle development, epithelial–mesenchymal transition (EMT), metabolism of tumors, and white adipose tissue [36,37,38,39]. Recurrent autosomal dominant mutations in the DNA binding domain of the *TWIST2* gene lead to two rare congenital abnormality disorders, Barber–Say syndrome (BSS) and ablepharon–macrostomia syndrome (AMS), which are expressed as dysmorphic facial characteristics and congenital malformations caused by ectodermal dysplasias [40]. In an animal model, Sonic et al. found that mice with *TWIST2* gene deletion failed to thrive within three days after birth due to glycogen stores dysfunction and proinflammatory cytokine elevation [41]. Pb could be deposited in bone for a long time; thus, TWIST may inhibit osteoblast maturation, which might relate to bone Pb deposition and calcium metabolism, and furthermore, may be related to HDL-C.

In addition, by improving hepatic steatosis, inflammation, and oxidative stress, Zhou et al. demonstrated the modulation profile of *TWIST2* in maintaining hepatic homeostasis [42]. The association between the lipid profile and the *TWIST2* gene family, including genetic polymorphisms of rs150813626, has not been reported in humans before. Compared to the rs150813626 GG carriers, our study showed the AA and AG carriers were significantly and negatively associated with HDL-C, which implies that the GG allele might be a protective genotype in Pb exposure. Although there have still been relatively few studies on the function of TWIST2, we found that the rs150813626 SNP may play a certain role in the relationship between blood lead level and HDL-C. However, the possible mechanism of rs150813626 SNP and the consequential function of the SNP require further investigation.

Another associated locus, rs1983079, is located on Chromosome 3, 3:114243087 (GRCh38) [43]. Some previous code names, rs58797957 and rs56490881, were merged as rs1983079 [44]. No gene consequence nor clinical significance were reported before [45]. In our study, compared to the rs1983079 GG carriers, the rs1983079 AA carriers and rs1983079 AG carriers had significantly higher HDL-C, at 8.028 mg/dL and 2.92 mg/dL, respectively, which implies that the rs1983079 AA carriers and AG carriers have a protective role in Pb exposure.

There are several limitations to the study. First, the study lacked information on the environmental exposure conditions of the participants. Instead, we used the blood Pb concentrations as the representative internal dose to represent the exposure conditions. Second, from this cross-sectional study, we could not explain the causal link between blood Pb and HDL-C. A further longitudinal study for clarifying the causality is required. Third, the expressions of these novel associated loci and the message translation from these SNPs are still unclear. Further functional studies of these SNPs are essential for understanding the influence of the association between Pb and lipid metabolism. Fourth, we lacked a questionnaire about the diet and medical records of the participants, which may be potential confounders of the study.

## 5. Conclusions

In the study, we demonstrated the first genome-wide evidence in a Taiwanese population that two associated SNPs, rs150813626, and rs1983079, were significantly associated with HDL-C in Pb-exposed workers. Furthermore, the interaction of rs150813626 SNPs and blood Pb levels influenced the relationships among HDL-C, blood Pb levels, and rs150813626 genetic polymorphisms. However, how the functional expression of these SNPs affects the relationship between Pb and lipid metabolism requires further investigation.

## Figures and Tables

**Figure 1 ijerph-19-01352-f001:**
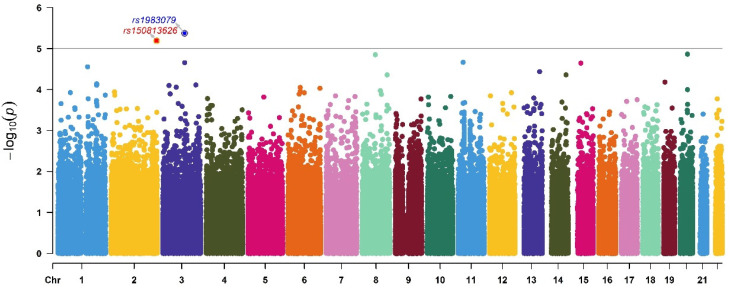
The Manhattan plots of the genome-wide association study of HDL-C in Pb-exposed workers. Two SNPs were found—rs150813626 (single-nucleotide variation in the TWIST2 gene on chromosome 2), and rs1983079 (unclear SNP on chromosome 3).

**Figure 2 ijerph-19-01352-f002:**
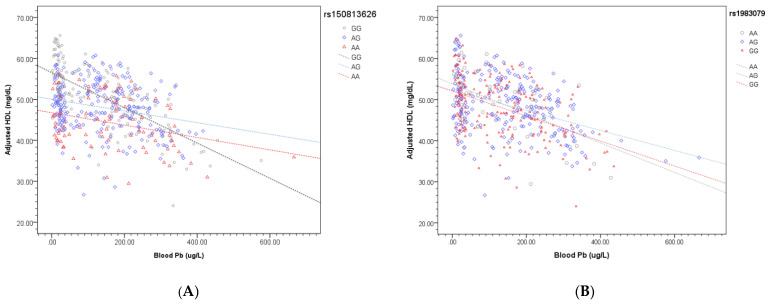
(**A**). The interaction plot for the regression of the association between HDL-C and blood Pb level according to rs150813626 single nucleotide polymorphisms, GG, AG, and AA carriers. (**B**). The interaction plot for the regression of the association between HDL-C and blood Pb level according to rs1983079 single nucleotide polymorphisms, AA, AG, and GG carriers.

**Table 1 ijerph-19-01352-t001:** Descriptive analysis of the demographic characteristics, blood Pb level, and lipid biomarkers.

	Mean ± SD/Number (%)	Medium	IQR
Age (year)	42.79 ± 10.48	42.42	34.25–51.85
Body Weight (kg)	66.73 ± 14.15	65.50	56.40–74.40
Body Height (cm)	163.33 ± 9.30	164.00	156.10–170.60
BMI (kg/m^2^)	24.90 ± 4.24	24.40	22.30–26.90
Blood Pb Level (ug/L)	139.94 ± 111.11	128.00	30.48–215.00
TG (mg/dL)	137.95 ± 144.02	99.00	67.00–168.00
TC (mg/dL)	205.31 ± 38.38	203.00	178.00–227.00
HDL-C (mg/dL)	48.45 ± 13.51	47.00	38.00–57.00
LDL-C (mg/dL)	132.68 ± 32.98	130.00	109.00–154.00
Gender			
Female	230 (45.0%)		
Male	281 (55.0%)		
Smoking			
Yes	166 (32.5%)		
No	345 (67.5%)		
Alcohol			
Yes	163 (31.9)		
No	332 (65.0)		

Abbreviations: BMI, body mass index; TG, triglyceride; TC, total cholesterol; HDL-C, high-density lipoprotein cholesterol; LDL-C, low-density lipoprotein cholesterol; IQR: interquartile range.

**Table 2 ijerph-19-01352-t002:** The regression model of TG, TC, HDL-C, LDL-C, blood Pb level, and other potential confounders in Pb-exposed workers.

	TG		TC		HDL-C		LDL-C	
	β (SE)	*p*	β (SE)	*p*	β (SE)	*p*	β (SE)	*p*
Blood Pb (ug/L)	−0.0061 (0.062)	0.93	0.019 (0.016)	0.25	−0.025 (0.005)	<0.001	0.010 (0.014)	0.50
Gender (male/female)	31.65 (14.03)	0.03	−9.89 (3.70)	0.01	−6.04 (1.23)	<0.001	−5.90 (3.27)	0.07
Age (year)	1.10 (0.63)	0.08	0.83 (0.17)	<0.001	0.001 (0.055)	0.99	0.62 (0.15)	<0.001
BMI (kg/m^2^)	7.65 (1.48)	<0.001	0.94 (0.39)	0.02	−1.004 (0.13)	<0.001	0.57 (0.35)	0.10
Smoking (yes/no)	54.57 (15.19)	<0.001	9.85 (4.00)	0.01	−2.11 (1.33)	0.11	3.22 (3.54)	0.36
Alcohol (yes/no)	18.91 (14.72)	0.20	7.50 (3.88)	0.05	−0.39 (1.29)	0.77	2.14 (3.43)	0.53
Constant	−139.80 (47.95)	<0.001	143.80 (12.64)	<0.001	81.20 (4.19)	<0.001	92.03 (11.19)	<0.001

Abbreviations: BMI, body mass index; TG, triglyceride; TC, total cholesterol; HDL-C, high-density lipoprotein cholesterol; LDL-C, low-density lipoprotein cholesterol.

**Table 3 ijerph-19-01352-t003:** The regression model of HDL-C, the three SNPs, other potential confounders, and the interaction of rs150813626 SNP and blood Pb in Pb-exposed workers.

	Model 1		Model 2		Model 3	
	β (SE)	*p*	β (SE)	*p*	β (SE)	*p*
Blood Pb (ug/L)	−0.023 (0.005)	<0.001	−0.036 (0.008)	<0.001	−0.026 (0.005)	<0.001
Gender (male/female)	−6.032 (1.22)	<0.001	−5.85 (1.22)	<0.001	−6.34 (1.20)	<0.001
Age (year)	0.018 (0.055)	0.746	0.022 (0.055)	0.682	0.002 (0.054)	0.972
BMI (kg/m^2^)	−1.014 (0.13)	<0.001	−1.008 (0.13)	<0.001	−0.94 (0.13)	<0.001
Smoking (yes/no)	−2.078 (1.32)	0.115	−2.18 (1.32)	0.098	−2.02 (1.30)	0.121
Alcohol (yes/no)	−0.46 (1.28)	0.721	−0.29 (1.28)	0.820	−0.62 (1.26)	0.624
rs150813626						
AA vs. GG	−6.34 (1.56)	<0.001	−10.27 (2.47)	<0.001		
AG vs. GG	−2.44 (1.20)	0.042	−4.90 (1.89)	0.010		
rs150813626 AA * BPb			0.027 (0.013)	0.036		
rs150813626 AG * BPb			0.018 (0.011)	0.091		
rs1983079						
AA vs. GG					8.028 (1.67)	<0.001
AG vs. GG					2.92 (1.16)	0.012
Constant	82.68 (4.16)	<0.001	83.89 (4.18)	<0.001	77.49 (4.24)	<0.001

BMI, body mass index; BPb, blood Pb level; Model 1, the regression model of HDL-C, rs150813626 SNPs, and other potential confounders; Model 2, the regression model of HDL-C, rs150813626 SNPs, other potential confounders, and the interaction of rs150813626 SNP and blood Pb level; Model 3, the regression model of HDL-C, rs1983079 SNPs, and other potential confounders.

## Data Availability

Data application https://taiwanview.twbiobank.org.tw (accessed on 1 December 2021).

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
