# Peer review of "Single Nucleotide Polymorphism of TWIST2 May Be a Modifier for the Association between High-Density Lipoprotein Cholesterol and Blood Lead (Pb) Level"

_ijerph, 2022, doi:10.3390/ijerph19031352_

Round 1

Reviewer 1 Report

In the manuscript titled Single Nucleotide Polymorphism of TWIST2 Maybe a Modifier for the Association of High-Density Lipoprotein-Cholesterol and Blood Pb, Yang et al presented results from SNP variations analysis on Taiwan biobank Chip version 2 for factory workers. They found a few significant SNPs and implicated TWIST2 for the Pb exposure.

For their GWAS analysis in their cohort they do not specify control population, neither they did any ethnicity analysis. Finding SNPs is this kind of small-scale analysis does not reflect any causal effect of these SNPs. Authors should have at least compared their analysis with the existing GWAS studies for lipidemia. While collecting samples from the subjects did authors include any dietary questions or any medication on medical records? I find this study lacking in the rigor to correlate these SNP variations with Pb exposure and hence inconclusive.

Reviewer 2 Report

The comments are attached 

Reviewer 3 Report

The paper entitled "Single Nucleotide Polymorphism of TWIST2 Maybe a Modifier for the Association of High-Density Lipoprotein-Cholesterol and Blood Pb" is well structured and presented.

I suggest that the authors review the English language of the document.

I also recommend that the authors reflect about the limitations of the study, presented in the discussion, and include some information in the conclusion in order to avoid misunderstandings, taking into account the need of expression and functional studies to understand the influence of the association between Pb and lipid metabolism.

Round 2

Reviewer 1 Report

I am still not convinced with the implication of TWIST2 in this study. The correlation is far from functional. Therefore, suggest to tone it down.

Authors should still describe or provide visuals about where in the TWIST2 locus the SNPs were found whether in its coding region or non-coding region how far from the TWIST2 gene.

Reviewer 2 Report

Comments on the Revised manuscript IJERPH-1514263

The authors have addressed the major concern by incorporating the required addition and modification in the revised manuscript. At this point the authors have significantly upgraded the manuscript and I’d be happy to recommend accepting the paper with following the completion of editorial requirements
